# Effect of Hofmeister Ions on Transport Properties of Aqueous Solutions of Sodium Hyaluronate

**DOI:** 10.3390/ijms22041932

**Published:** 2021-02-16

**Authors:** Lenka Musilová, Aleš Mráček, Věra Kašpárková, Antonín Minařík, Artur J. M. Valente, Eduarda F. G. Azevedo, Luis M. P. Veríssimo, M. Melia Rodrigo, Miguel A. Esteso, Ana C. F. Ribeiro

**Affiliations:** 1Department of Physics and Materials Engineering, Faculty of Technology, Tomas Bata University in Zlín, Vavrečkova 275, 760 01 Zlín, Czech Republic; lmusilova@utb.cz (L.M.); minarik@utb.cz (A.M.); 2Centre of Polymer Systems, Tomas Bata University in Zlín, tř. Tomáše Bati 5678, 760 01 Zlín, Czech Republic; vkasparkova@utb.cz; 3Department of Fat, Surfactant and Cosmetics Technology, Faculty of Technology, Tomas Bata University in Zlín, Vavrečkova 275, 762 72 Zlín, Czech Republic; 4Department of Chemistry, University of Coimbra, CQC, 3004-535 Coimbra, Portugal; edy.gil.azevedo@gmail.com (E.F.G.A.); verissimo@uc.pt (L.M.P.V.); mmelia.rodrigo@uah.es (M.M.R.); anacfrib@ci.uc.pt (A.C.F.R.); 5U.D. Química Física, Universidad de Alcalá, 28871 Alcalá de Henares, Spain; mangel.esteso@ucavila.es; 6Universidad Católica Santa Teresa de Jesús de Ávila, Calle los Canteros s/n, 05005 Ávila, Spain

**Keywords:** sodium hyaluronate, transport properties, viscosity, Hofmeister series

## Abstract

Tracer diffusion coefficients obtained from the Taylor dispersion technique at 25.0 °C were measured to study the influence of sodium, ammonium and magnesium salts at 0.01 and 0.1 mol dm^−3^ on the transport behavior of sodium hyaluronate (NaHy, 0.1%). The selection of these salts was based on their position in Hofmeister series, which describe the specific influence of different ions (cations and anions) on some physicochemical properties of a system that can be interpreted as a salting-in or salting-out effect. In our case, in general, an increase in the ionic strength (i.e., concentrations at 0.01 mol dm^−3^) led to a significant decrease in the limiting diffusion coefficient of the NaHy 0.1%, indicating, in those circumstances, the presence of salting-in effects. However, the opposite effect (salting-out) was verified with the increase in concentration of some salts, mainly for NH_4_SCN at 0.1 mol dm^−3^. In this particular salt, the cation is weakly hydrated and, consequently, its presence does not favor interactions between NaHy and water molecules, promoting, in those circumstances, less resistance to the movement of NaHy and thus to the increase of its diffusion (19%). These data, complemented by viscosity measurements, permit us to have a better understanding about the effect of these salts on the transport behaviour of NaHy.

## 1. Introduction

Hyaluronan (sodium salt of hyaluronic acid, NaHy), was firstly obtained by Meyer et al. [1] from the vitreous humor of cattle eyes. It consists of a disaccharide repeating sequence of d-glucuronic acid and *N*-acetyl-d-glucosamine, linked via alternating β-(1→4) and β-(1→3) glycosidic bonds (Figure 1) [2]. Hyaluronic acid (HyA) is a natural linear polysaccharide, being naturally present and abundant in all biologic fluids of some bacteria and all vertebrates [2,3,4]. Due to its biocompatibility, biodegradability and presence in the native extracellular matrix of tissues [5], HyA has been extensively applied for medical [6,7] and biomedical applications [8,9], including drug delivery [10,11,12], bioprinting [13] and tissue engineering [14], cosmetics [15] and wastewater treatment [16].

For most of these applications, knowledge of the transport properties of HyA in aqueous solutions plays a key role. For example, whilst for electrospinning the solution viscosity is relevant, for drug delivery the understanding of mass transport by diffusion is essential as a limiting kinetic constant. Despite this, only few articles describe the behaviour of hyaluronic acid solutions in terms of transport properties. Verissimo et al. [17] reported mutual diffusion coefficients of sodium hyaluronate in the concentration range 0.50 to 50 g dm^−3^, at 25 °C, allowing the estimation of different parameters such as the diffusion coefficient at infinitesimal concentration, the limiting ionic conductivity and the tracer diffusion coefficient of hyaluronate ion. Later, these studies were complemented by viscosity data, and its influence on the diffusion coefficients were discussed [18]. However, only few data have shown the effect of the ionic strength on the HyA behavior in aqueous solution. By using optical techniques, Wik and Comper studied the effect of the ionic strength on the mutual diffusion coefficient of hyaluronate. However, the obtained results must be considered as approximate according to the authors [19]. Some of us [20] have reported dynamic light-scattering (DLS), viscosity and surface tension (SFT) measurements to characterize the influence of the following salts: Na_2_SO_4_, (NH_4_)_2_SO_4_, NaSCN, NH_4_SCN and NaCl, on the behavior of hyaluronan in diluted solutions at a temperature range of 15–45 °C. In the sequence of this work, we intend to extend these studies measuring viscosities for the same aqueous systems but using other NaHy with lower molecular mass (Mw = 124 kDa), and also characterise the influence of these particular salts on the diffusion of NaHy using the same samples of this compound (i.e., Mw = 1.8–2.1 MDa and Mw = 124 kDa). The selection of these salts was based on the ordering of Hofmeister series [21] and its related properties (viscosity or surface energy and entropies of ion solvation) [22], this classification being proposed in accordance with the abilities of these ions to induce the water structuring or breaking (Figure 2). The ions on the left side are called kosmotropes (“water structure makers”), while on the right side are chaotropes (“water structure breakers”) [21]. It is known that Hofmeister or specific ion effects affect some physicochemical properties of aqueous [23,24] and nonaqueous [25,26] systems. As an example, there are studies made by Mráček et al. [27] about the influence of Hofmeister series ions on HyA behaviour and HyA film-swelling. Additionally diffusion coefficients of water on the swelling process of films of sodium hyaluronate and their hydrophobically modified derivatives have also been studied [28,29]. Mráček et al. [29] also showed, by using viscosity measurements, that the presence of specific ions such as sulphate and thiocyanate anions, affects the behaviour of NaHy in aqueous solutions due to its ability to act as “structure making” and “structure breaking”, leading to the expansion or contraction of the NaHy coil.

Thus, the present work focuses on the investigation of the effect of different salts (Na_2_SO_4_, (NH_4_)_2_SO_4_, MgSO_4_, NaSCN, NH_4_SCN, NaCl and LiCl), with different kosmotropic/chaotropic [30,31], on the mutual interdiffusion coefficients and viscosities of NaHy, at 25 °C.

## 2. Results and Discussion

### 2.1. Apparent Tracer Diffusion Coefficients of Sodium Hyaluronate

Table 1 and Table 2 present the apparent tracer experimental diffusion coefficients, ^app^*D*^0^_1T_, for aqueous pseudo binary systems containing hyaluronate (Mw = 1.8–2.1 MDa and Mw = 124 kDa) and some salts at 0.01 or 0.1 mol dm^−3^ at 25 °C. These ^app^*D*^0^_1T_ values were obtained from at least three independent runs. The uncertainty of these values is not greater than 3%.

From Table 1, it can be observed, in general, that almost all added salts at 0.01 mol dm^−3^ contributed to the decrease of the diffusion of NaHy (Mw = 1.8–2.1 MDa). In fact, the deviations between the tracer diffusion coefficient values of NaHy in these supporting electrolytes and the limiting diffusion coefficient of the NaHy in water, at the same temperature, are negative (i.e., ^app^*D*^0^_1T_ < *D*^0^_NaHy_). However, these deviations were more significant for systems containing salts at 0.01 mol dm^−3^. In these cases, these negative deviations (at most, −54% for MgSO_4_), indicate the presence of salting-in effects.

A possible explanation for this phenomenon can be given if we consider that the Mg^2+^ ions are classified as poorly hydrated kosmotropic cations (Figure 2, and thus promote the salting-in effect, contributing to the contraction of NaHy chain and inducing a significant increasing of its chain stiffness. In these circumstances, there is the possibility to have a folded structure involving intramolecular hydrogen bonds. Similar behaviour is also observed in salts containing thiocyanates at 0.01 mol dm^−3^, as well as chlorides at same concentration. 

Support for these facts is shown in previous viscosity studies for the same systems [20], that is, NaHy in water and in saline solutions. It was concluded that the absence of salts leads to the increase of the repulsions between the unities of sodium hyaluronate, and consequently to the increase of the hydrodynamic volume of the coil of NaHy and, of course, of its viscosity. However, at high ionic strength, the charges due to the carboxylate groups on the NaHy chain are completely screened, thus diminishing the repulsion between them, which hinders the expansion of the coil.

The entities of NaHy offer more frictional resistance to motion through the liquid and, consequently, the diffusion coefficient of this aqueous system becomes smaller.

Concerning the other sulphates, (NH_4_)_2_SO_4_, and Na_2_SO_4_, we can verify the deviations are less accentuated (that is, −16.8% and −35.5%). This is not a surprise if we take in account that NH_4_^+^ (chaotropic) and Na^+^ (border line between chaotropic and kosmotropic) cations are considered hydrated.

However, some added salts at *c* = 0.1 mol dm^−3^, with special relevance for NH_4_SCN, increased the value of the diffusion coefficient ^app^*D*^0^_1T_ of the NaHy from 6 to 20% (this maximum value was observed with NH_4_SCN, indicating the presence of salting-out effects. NH_4_^+^ being a chaotropic cation, will favour interactions between NaHy and water, and, consequently, the entities of NaHy offer less frictional resistance to motion through the liquid. Thus, the diffusion coefficient of this aqueous system becomes larger. 

Relative to the study of the effect of the same salts on diffusion of NaHy but having a lower molecular mass (Table 2), it was observed that, in general, the deviations between the apparent tracer diffusion coefficients of NaHy in aqueous electrolytic solutions at 0.01 or 0.1 mol dm^−3^, and the limiting diffusion coefficient of the aqueous solutions of NaHy at the same temperature were positive. In those circumstances, we can say that NaHy with lower molecular mass has less unities of NaHy in water and, consequently, has less interactions between these entities and water molecules, thus, leading to the increased diffusion coefficient.

### 2.2. Viscosity

Viscosimetry is an efficient method suitable for determining changes in the conformation of polymers in solutions. Using this method, viscosity of diluted polymer solutions can be expressed by limiting viscosity number [*η*] commonly calculated according to the Huggins equation through extrapolation of reduced viscosity *η*_red_ versus concentration dependence to zero concentration. However, interesting information can be obtained about the behaviour of polymers from the course of the viscosity curves, mainly in the case when the studied polymer belongs to the polyelectrolyte group, which true for NaHy. Therefore, in this work, we evaluated viscosity data for NaHy with Mw = 124 kDa from the following two points of view: (1) we compared values of *η*_red_ for NaHy in the aqueous salt solutions determined at the highest polymer concentration of 1.5 g L^−1^; and (2) we evaluated the course of the *η*_red_ versus *c* dependence. Here, it is mainly interesting to look into polymer behaviour at the lowest concentrations (0.1 to 0.5 g L^−1^). 

We assessed the values of reduced viscosity *η*_red_ at the concentration of 1.5 g L^−1^ in presence of studied salts and they decreased in the following order: Na_2_SO_4_ > (NH_4_)_2_SO_4_ > NaSCN > LiCl > NH_4_SCN > NaCl > MgSO_4_ (Figure 3). Looking at the order, it is seen that the behaviour of sulfates with monovalent cations Na^+^ or NH_4_^+^ differs from sulfate with divalent Mg^2+^, which obviously illustrates that the type of cation present in the solution affects the NaHy conformation. Indeed, in the presence of salts, part of the ionized groups on the polymer chain can be neutralized and the polymer can form the more compact conformation. This effect is controlled by the type of counterion in the solution, and the association of polymer coil diminishes in the order roughly following the Hofmeister series, with divalent cations being more efficient in comparison with monovalent ones causing bigger coil shrinkage [32]. However, influence of anions on coil expansion can also be noticed. At this highest polymer concentration, the *η*_red_ values are higher for NaHy in sulfates with monovalent cations (Na_2_SO_4_, (NH_4_)_2_SO_4_) than *η*_red_ for NaHy in both thioisocyanates (NaSCN, NH_4_SCN). This shows the smaller expansion of the NaHy coil in the presence of the SCN^−^. Here, the described behaviour of NaHy conforms with findings reported in [20] for NaHy with notably higher molecular mass of 1.8–2.1 MDa.

The course of the *η*_red_ vs. *c* dependences depicted in Figure 3 affords additional interesting information on the studied samples and proves that NaHy behaves as typical polyelectrolyte polymer. This type of behaviour is commonly characterized by a nonlinear course of the *η*_red_ vs. *c* plot with upward curvature at the lowest NaHy concentrations, recorded mainly for NaHy dissolved in demineralised water (Figure 4).

If we look at Figure 3 in more detail, the *η*_red_ of all NaHy solutions decrease linearly with decreasing polymer concentration down to ≈0.5 g L^−1^ with a slope depending on the type of the dissolved salt. However, at the lowest concentrations 0.25 to 0.1 g L^−1^, some of the salts (NaSCN, Na_2_SO_4_, MgSO_4_, LiCl) induce the upwards curvature of the viscosity curve similar to NaHy in water. In contrast, *η*_red_ vs. *c* dependences for (NH_4_)_2_SO_4_, NH_4_SCN and NaCl curve downwards. If we accept the prerequisite that the NaCl behaviour is exceptional due to the “borderline” position of both anion and cation in Hofmeister series, both these salts (NH_4_)_2_SO_4_, NH_4_SCN have it in common that a chaotropic cation is present. On the contrary, the upwards curvature can be observed for NaHy dissolved in salts containing kosmotropic Mg^2+^ and Na^+^, which is also classified among the kosmotropes, though it is positioned at the “borderline”. At these two lowest concentrations, the NaHy is highly diluted and the effect of dissolved salts begins to predominate over the effect of polymer. The polymer-polymer interactions are notably reduced relatively to polymer-solvent interactions, which prevail. This type of the salt, which can interact both with polymer and water molecules, starts to impact more on the polymer viscosity, which is also visible at the graph curves. At the lowest NaHy concentrations, the salts or more specifically their cations and anions, influence the structure of water in terms of “structure making” or “structure breaking”. In this respect it can be emphasized that chaotropic ions that disrupt the structure of bulk water favour polymer shrinkage as the entropic difference between bound water and bulk water increases [33]. 

In this respect, it is interesting to look into the viscosity of aqueous solutions of salts in the absence of the polymer. Here the viscosity increases in the following order NH_4_SCN < H_2_O < (NH_4_)_2_SO_4_ < NaSCN < NaCl < Na_2_SO_4_ < LiCl < MgSO_4_. Relatively to the viscosity of demineralized water, only NH_4_SCN the salt with both chaotropic cation and anion (CH_C_CH_A_), decreases the viscosity of water. In presence of all other tested salts, whether they are combination of CH_C_K_A_, K_C_CH_A_ or K_C_K_A_, the viscosity of water with salt increases relatively to that of demineralised water. According to Applebey [34], the effect of water dissolved ions on viscosity can be divided into two main effects: 1) decrease in viscosity caused by the breakdown of molecular clusters of water, (H_2_O)_3_, to simple molecules, and 2) increase in viscosity caused by the presence of salt ions and nonionized salt molecules. In the case of ions that are not fully hydrated, such as Cs^+^, K^+^, NH_4_^+^, I^−^, Br^−^, NO_3_^−^, the decrease in viscosity is likely caused by the breakdown of clusters of water molecules. However, as most ions are fully hydrated, they are expected to increase the viscosity of water. 

Table 3 compares the effect of the salts on the NaHy behaviour evaluated from diffusion coefficients and viscosity data. The values of apparent tracer diffusion coefficients ^app^*D*^0^_IT_ of NaHy with Mw = 124 kDa, determined in presence of salts with ionic strength 0.1 mol L^−1^ at 25 °C, are sorted by the increasing values corresponding with values of *η*_red_ determined for NaHy at the lowest and the highest polymer concentrations and viscosity of aqueous salt solutions in the absence of NaHy. Here, it is obvious that the two used methods describe the NaHy behaviour in aqueous salt solutions by different ways. When we sort the ^app^*D*^0^_IT_ according to increasing values, the order is as follows: MgSO_4_ < Na_2_SO_4_ < LiCl < NaSCN < NaCl < (NH_4_)_2_SO_4_ < NH_4_SCN. It follows that on the left side there are the salts with kosmotropic cation and anion (K_C_K_A_) followed by salts containing K_C_CH_A_ and CH_C_K_A,_ and on the right side by salts with chaotropic cation and anion (CH_C_CH_A_). However, the viscosity values behave differently, as already discussed above. Surprisingly, the data from diffusion measurements are in reasonably good agreement (opposite order) with reduced viscosity values recorded for water with dissolved salts, where NaHy was absent.

## 3. Materials and Methods

### 3.1. Materials

Sodium hyaluronate (NaHy), Mw = 1.8–2.1 MDa and Mw = 124 kDa were a kind gift of Contipro Ltd., (Dolní Dobrouč, Czech Republic). Sodium sulfate (Na_2_SO_4_), ammonium sulfate ((NH_4_)_2_SO_4_), sodium thiocyanate (NaSCN), ammonium thiocyanate (NH_4_SCN), sodium chloride (NaCl), lithium chloride (LiCl) and magnesium sulfate (MgSO_4_) were purchased from Sigma-Aldrich. All salts used were received at a purity higher than 97.5% and used as delivered. The solutions for the diffusion measurements were prepared in calibrated volumetric flasks using also Milli-Q water (from A10 Millipore) and were freshly prepared.

### 3.2. Sample Preparation for Viscosity Measuring 

For the viscosity measurements, the hyaluronan solution was prepared in concentration of 1.5 g L^−1^ by dissolving NaHy powder in prepared salt solutions (ionic strength of 0.1 mol L^−1^) under continuous stirring, followed by 24-h dissolving at 50 °C. This concentration of NaHy was then diluted with respective aqueous salt solution (MgSO_4_, Na_2_SO_4_, (NH_4_)_2_SO_4_, NaSCN, NH_4_SCN, NaCl, LiCl) to obtain NaHy solutions with concentrations 1.0, 0.7; 0.5; 0.25 and 0.1 g L^−1^.

Viscosity measurements were performed using an automated viscometer SI Viscoclock (Schott Instruments, Karlsbad, Germany) equipped with Ubbelohde capillary viscometers at 25.0 ± 0.1 °C. The mean flow time of NaHy solutions though the capillary was calculated from five repeatable measurements. The relative *η*_rel_, specific *η*_sp_ and reduced *η*_red_ viscosities of NaHy samples were subsequently calculated. The dependences between the reduced viscosity and concentration according to Huggins were then plotted and compared.

### 3.3. Measurements of Diffusion Coefficients Using Taylor Technique

#### 3.3.1. Brief Description about Some Concepts on Diffusion

In binary NaHy^+^ H_2_O solutions, the Na^+^ and Hy^−^ ions are required by electroneutrality to diffuse at the same speed. As a result, mutual diffusion of the electrolyte is described by Fick’s law [35,36] with a single binary diffusion coefficient *D*, which is a weighted average of the diffusion coefficients of the ionic species.
(1)J(NaHy)=−D∇C

From Nernst’s equation, the limiting diffusion coefficient of a sodium hyaluronate (NaHy) in terms of the diffusion coefficients of the Na^+^ and the hyaluronate anions in units of sodium hyaluronate Hy^−^ ions is:(2)$$D^{0}=\frac{2D_{{\mathrm{Na}}^{+}}^{0}D_{{\mathrm{HY}}^{-}}^{0}}{D_{{\mathrm{Na}}^{+}}^{0}{+D}_{{\mathrm{HY}}^{-}}^{0}}$$

However, a higher value would be expected for the tracer diffusion coefficient, *D*^0^, compared to what would be predicted, having in mind that *D*^0^_HY_− is not the limiting diffusion of polymeric HyA-anions. 

The mutual diffusion of NaHy in solutions of a supporting electrolyte, such as aqueous Na_2_SO_4_, differs qualitatively from its diffusion in pure water. For example, the Na^+^ and SO_4_^2−^ ions are not required to diffuse at the same speed. The diffusion of the electrolytes is coupled by the electric field (diffusion potential) generated by concentration gradients of ions with different mobilities. Diffusion in ternary NaHy(1) + salt(2) + H_2_O solutions is, therefore, described by the coupled Fick equations [37,38].
(3)$$J_{1}\left(\mathrm{NaHy}\right)=-D_{11}\nabla C_{1}-D_{12}\nabla C_{2}$$
(4)$$J_{2}\left(\mathrm{salt}\right)=-D_{21}\nabla C_{1}-D_{22}\nabla C_{2}$$

*J*_1_, *J*_2_, and *C*_1_, *C*_2_ are, respectively, the molar fluxes and the gradients in the concentrations of NaHy (solute 1) and salt (solute 2). *D*_11_, *D*_12_, *D*_21_ and *D*_22_ are the ternary mutual diffusion coefficients. The main diffusion coefficients, *D*_11_ and *D*_22_, give the flux of each solute (NaHy and salt, respectively) produced by its own concentration gradient. Cross-diffusion coefficients *D*_12_ and *D*_21_ are included for the flux of NaHy caused by the salt concentration gradient (*C*_2_) and the flux of salt caused by the NaHy concentration gradient (∇*C*_1_). 

In a particular case, such as *C*_1_/*C*_2_ → 0 (corresponding to a large molar excess of salt relative to NaHy), the limiting value of the main coefficient *D*_11_, of the NaHy component changes from the Nernst value in pure water (Equation (2)) to the tracer diffusion coefficient value, *D*_HY−_^0^, of the Hy^−^ ion in aqueous salt solution. Also, in the limit *C*_1_/*C*_2_ → 0, the cross-coefficient *D*_12_ drops to zero because a salt (2) concentration gradient is unable to drive a coupled flow of NaHy (1) in a Hy^−^ free solution. The cross-coefficient *D*_21_, however, is not necessarily zero in the limit *C*_1_/*C*_2_ → 0 and, in fact, it can be quite large, especially for mixed electrolyte solutes.

Based on these considerations, the tracer diffusion of NaHy in supporting salt solutions is described by the equations:(5)$$J_{1}\left(\mathrm{NaHy}\right)=-D_{11}\nabla C_{1}$$
(6)$$J_{2}\left(\mathrm{salt}\right)=-D_{21}\nabla C_{1}-D_{22}\nabla C_{2}$$

Because *D*_21_ is not zero, a concentration gradient in the NaHy tracer will drive a significant coupled flow of NaCl in the carrier solution, producing ternary Taylor peaks (two overlapping Gaussian profiles).

#### 3.3.2. The Taylor Technique: Binary Diffusion

The theory of the Taylor dispersion technique is well described in the literature [35,37,38,39] and consequently the authors only point out some relevant points concerning such a method on the experimental determination of binary, ternary and tracer diffusion coefficients, respectively. 

Dispersion methods for diffusion measurements are based on the dispersion of small amounts of solution injected into laminar carrier streams of solvent or solution of different composition flowing through a long capillary tube. The length of the Teflon dispersion tube used in the present study was measured directly by stretching the tube in a large hall and using two high quality theodolites and appropriate mirrors to accurately focus on the tube ends. This technique gave a tube length of 3.2799 (± 0.0001) × 10^3^ cm, in agreement with less-precise check measurements using a good-quality measuring tape. The radius of the tube, 0.05570 (±0.00003) cm, was calculated from the tube volume obtained by accurately weighing (resolution 0.1 mg) the tube when empty and when filled with distilled water of known density. 

At the start of each run, a 6-port Teflon injection valve (Rheodyne, model 5020, Merck KGaA, Darmstadt, Germany) was used to introduce 0.063 cm^3^ of solution into the laminar carrier stream of slightly different composition. A flow rate of 0.17 cm^3^ min^−1^ was maintained by a metering pump (Gilson model Minipuls 3, Middleton, WI, USA) to give retention times of about 8 × 10^3^ s. The dispersion tube and the injection valve were kept at 25 °C (±0.01 °C) in an air thermostat.

Dispersion of the injected samples was monitored using a differential refractometer (Waters model 2410) at the outlet of the dispersion tube. Detector voltages, *V*(*t*), were measured at accurately timed 5 s intervals with a digital voltmeter (Agilent 34401 A, Santa Clara, CA, USA) with an IEEE-488 interface. Binary diffusion coefficients were evaluated by fitting the dispersion equation:*V*(*t*) = *V*_0_ + *V*_1_*t* + *V*_max_ (*t*_R_/*t*)^1/2^ exp[−12*D*(*t* − *t*_R_)^2^/*r*^2^*t*](7)
to the detector voltages. The additional fitting parameters were the mean sample retention time *t*_R_, peak height *V*_max_, baseline voltage *V*_0_, and baseline slope *V*_1_.

#### 3.3.3. The Taylor Technique: Ternary Diffusion

The NaHy/salt system should be considered a ternary system, and we are actually measuring the tracer diffusion coefficients *D*^0^_11_ but not *D*^0^_12_, *D*^0^_21_ and *D*^0^_22_. For ternary Taylor experiments, samples of solutions of composition ${\bar{C}}_{1}+\Delta C_{1}$, ${\bar{C}}_{2}+\Delta C_{2}$ are injected into laminar carrier streams of composition ${\bar{C}}_{1}$, ${\bar{C}}_{2}$. The injected solutes spread out as they flow through a long capillary tube. A high-precision differential refractive index detector at the tube outlet monitor the broadened distributions of the dispersed solutes. 

At time *t* after injection, the solute concentrations at the detector are [40]:(8)$$C_{1}\left(t\right)={\bar{C}}_{1}+\frac{2}{r^{3}u}\sqrt{\frac{3}{t\pi^{3}}}\left[A_{11}\sqrt{D_{1}}e^{-12D_{1}{\left(t-t_{R}\right)}^{2}/r^{2}t}+A_{12}\sqrt{D_{2}}e^{-12D_{1}{\left(t-t_{R}\right)}^{2}/r^{2}t}\right]$$
(9)$$C_{2}\left(t\right)={\bar{C}}_{2}+\frac{2}{r^{3}u}\sqrt{\frac{3}{t\pi^{3}}}\left[A_{21}\sqrt{D_{1}}e^{-12D_{1}{\left(t-t_{R}\right)}^{2}/r^{2}t}+A_{22}\sqrt{D_{2}}e^{-12D_{1}{\left(t-t_{R}\right)}^{2}/r^{2}t}\right]$$
in which *D*_1_ and *D*_2_ are the eigenvalues of the matrix of the ternary *D_ik_* diffusion coefficients. The detailed description of this treatment as well as the definition all parameters can be found in the literature [40].

#### 3.3.4. Tracer and Apparent Tracer Diffusion

A special case of ternary diffusion arises when one of the solutes is present in trace amounts [9,10,11]. In the limit *C*_1_/*C*_2_ → 0, for example, *D*_11_ is the tracer diffusion coefficient of solute 1 and *D*_22_ is the binary diffusion coefficient of solute 2 in the pure solvent. As noted in Section 3.3.1, *D*_12_ vanishes but *D*_21_ is not necessarily equal to zero. 

The tracer diffusion of solute 1 in solutions of solute 2 is measured by injecting small volumes of solution containing both solutes 1 and 2 into carrier solutions of pure solute 2. Strong dilution of solute 1 with the carrier solution ensures its tracer diffusion. Under these conditions, because *D*_12_ = 0, ${\bar{C}}_{1}$ = 0, *D*_1_ = *D*_11_ (the tracer diffusion coefficient of solute 1) and *D*_2_ = *D*_22_ (the binary diffusion coefficient of solute 2), the general expressions for ternary concentrations profiles (Equations (8) and (9)) simplify to:(10)$$C_{1}\left(t\right)=\frac{2\Delta V\Delta C_{1}}{r^{3}u}\sqrt{\frac{3D_{11}}{\pi^{3}t}}e^{-12D_{11}{\left(t-t_{R}\right)}^{2}/\left(r^{2}t\right)}$$
(11)$$C_{2}\left(t\right)={\bar{C}}_{2}+\frac{2\Delta V}{r^{3}u}\sqrt{\frac{3}{\pi^{3}t}}\left[\begin{array}{c} \frac{D_{21}\Delta C_{1}}{\left(D_{11}-D_{22}\right)}\sqrt{D_{11}}e^{-12D_{11}\frac{{\left(t-t_{R}\right)}^{2}}{r^{2}t}}+\\ \frac{(D_{11}-D_{22})\Delta C_{2}-D_{21}\Delta C_{1}}{\left(D_{11}-D_{22}\right)}\sqrt{D_{22}}e^{-12D_{22}{\left(t-t_{R}\right)}^{2}/r^{2}t} \end{array}\right]$$

Due to the coupled diffusion of solute 2 caused by the concentration gradient in solute 1, tracer dispersion profiles for solute 1 generally resemble two overlapping Gaussian peaks of variance $r^{2}t_{R}/\left(48D_{11}\right)$ and $r^{2}t_{R}/\left(48D_{22}\right)$.

In this work, however, we considered as an approach, that for three-component solution, water (0) + NaHy(1) + salt(2), the values of tracer diffusion coefficient for the NaHy, *D*_1_, can also be estimated by assuming there are no coupled diffusion, thus, treating the NaHy tracer Taylor peaks as simple binary ones and described by a single “pseudo-binary” tracer diffusion coefficient. In this case, we may perform a Taylor experiment for the tracer diffusion of solute 1 in which the tracer diffusion of NaHy (1) in aqueous salt (2) solutions is measured by injecting a solution of composition $C_{1}=\Delta C_{1},\;{\bar{C}}_{2}=C_{2}$ into a carrier solution of composition $C_{1}=0,\;{\bar{C}}_{2}=C_{2}$. In this case, Equations (12) and (13)
(12)$$C_{1}\left(t\right)=\frac{2\Delta V\Delta C_{1}}{r^{3}u}\sqrt{\frac{3D_{11}}{\pi^{3}t}}e^{-12D_{11}{\left(t-t_{R}\right)}^{2}/r^{2}t}$$
(13)$$C_{2}\left(t\right)={\bar{C}}_{2}+\frac{2\Delta V}{r^{3}u}\sqrt{\frac{3}{\pi^{3}t}}\left[\frac{D_{21}\Delta C_{1}}{\left(D_{11}-D_{22}\right)}\sqrt{D_{11}}e^{-12D_{11}{\left(t-t_{R}\right)}^{2}/r^{2}t}+\frac{(D_{11}-D_{22})\Delta C_{2}-D_{21}\Delta C_{1}}{\left(D_{11}-D_{22}\right)}\sqrt{D_{22}}e^{-12D_{22}{\left(t-t_{R}\right)}^{2}/r^{2}t}\right]$$
for the resulting concentration profiles *C*_1_(*t*) and *C*_2_(*t*) give
(14)$$S\left(t\right)=\bar{S}+\frac{2\Delta V\Delta C_{1}R_{1}}{r^{3}u}\sqrt{\frac{3}{\pi^{3}t}}\left[\left(1+\frac{R_{2}}{R_{1}}\frac{D_{21}}{D_{11}-D_{22}}\right)\sqrt{D_{11}}e^{-12D_{11}{\left(t-t_{R}\right)}^{2}/r^{2}t}\frac{R_{2}}{R_{1}}\frac{D_{21}}{D_{11}-D_{22}}\sqrt{D_{22}}e^{-12D_{22}{\left(t-t_{R}\right)}^{2}/r^{2}t}\right]$$
indicating the detector signal is the sum of overlapping Gaussians for diffusion coefficients *D*_11_ and *D*_22_ with relative weights {1 + (*R*_2_/*R*_1_)*D*_21_/(*D*_11_−*D*_22_)} and {−(*R*_2_/*R*_1_)*D*_21_/(*D*_11_−*D*_22_)}, respectively. Accordingly, the apparent diffusion coefficient obtained by treating the ternary tracer Taylor peak as a binary peak is: (15)Dapp1Tracer=(1+R2R1D21D11−D22)D11−(R2R1D21D11−D22)D22

Equation (15) can be significantly simplified to:(16)Dapp1Tracer=D11+R2R1D21
to obtain an apparent tracer diffusion coefficient, ^app^*D*_1Tracer_, for solute 1.

This pseudobinary treatment ignores the fact that tracer ionic diffusion in a supporting electrolyte is a multicomponent process involving coupled diffusion of the supporting electrolyte and, consequently, does not allow accurate prediction of tracer diffusion coefficients of NaHy. However, it is a useful approach, permitting us to provide qualitative information about the effect of the proposed salts on diffusion of NaHy, overcoming the great difficulties revealed in the measurement of diffusion coefficients of these systems, mainly motivated by the presence of fluids of very high viscosity.

Support for this is given by other studies [41], showing in these particular cases that the apparent tracer diffusion coefficients of these resorciararenes decrease as the concentration of the NaCl in the supporting electrolyte increases, approaching the true tracer diffusion coefficients obtained from experimental ternary Taylor dispersion profiles.

## 4. Conclusions

Based on these measurements of diffusion coefficients of systems containing sodium hyaluronate and electrolytes in aqueous solutions, assuming that in the present experimental conditions this system is pseudobinary, we conclude that the diffusion of this polysaccharide in aqueous solutions is strongly affected by the presence of the electrolytes. The behaviour of diffusion of NaHy in aqueous solutions changes in the presence of the salts as a result of the salting-in or salting-out effect. The salting-out effect is more favourable at high salt concentrations and when using samples of NaHy of lower molar mass. The structural differences for the forms of NaHy in solutions containing different electrolytes (extended and contracted structures, respectively) can be responsible for these phenomena. 

Diffusion coefficients measured for aqueous solutions of systems containing sodium hyaluronate and salts, provide transport data necessary to model the diffusion in pharmaceutical and engineering applications.

Changes in viscosity reflect polymer interactions with the respective salt in solution, which result in bigger or smaller shrinkage of the NaHy coil. In aqueous solutions, the salts dissociate, and cations/anions interact with functional groups present in NaHy chain dependent of their position in the Hofmeister series and polymer concentration (at given ionic strength). At the lowest-used polymer concentrations, the cations and anions influence NaHy viscosity through changing the structure of water in terms of “structure making” or “structure breaking” effects.

## Figures and Tables

**Figure 1 ijms-22-01932-f001:**
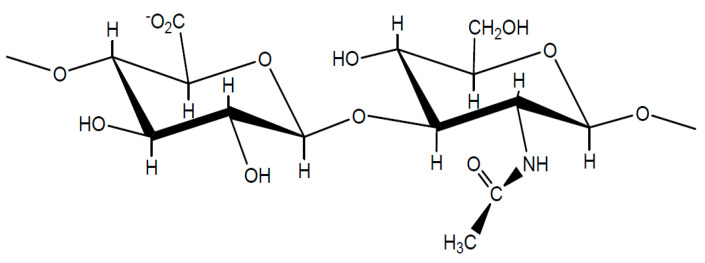
Molecular structure of the disaccharide repeating unit in hyaluronic acid salt.

**Figure 2 ijms-22-01932-f002:**
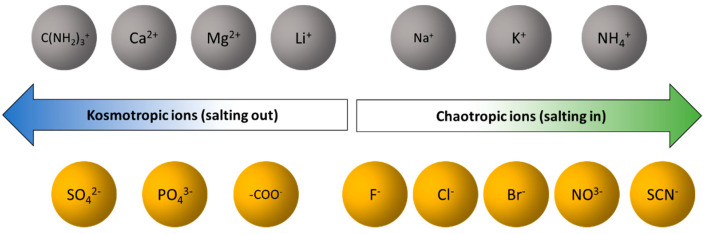
Typical ordering of Hofmeister ions series based on precipitation studies of solutions of typical and anionic proteins.

**Figure 3 ijms-22-01932-f003:**
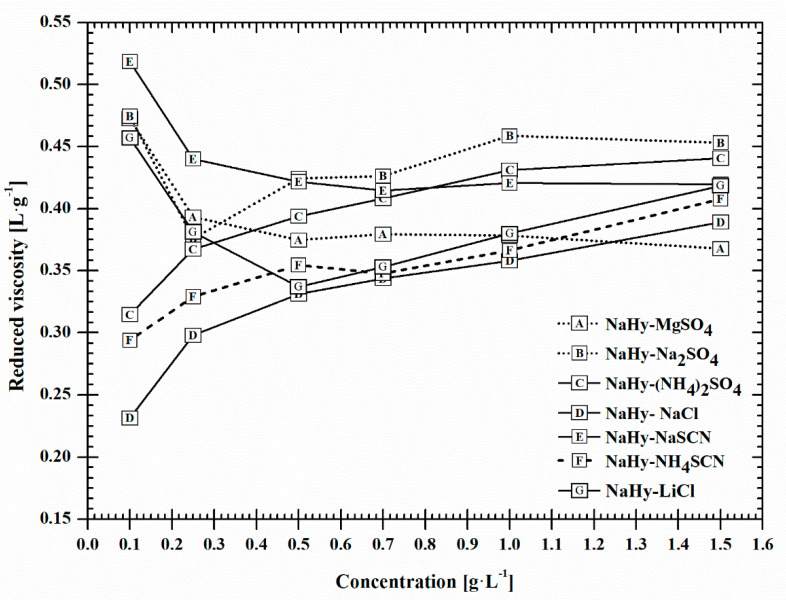
Dependence of reduced viscosity on concentration determined for NaHy dissolved in studies salts plotted according to Huggins.

**Figure 4 ijms-22-01932-f004:**
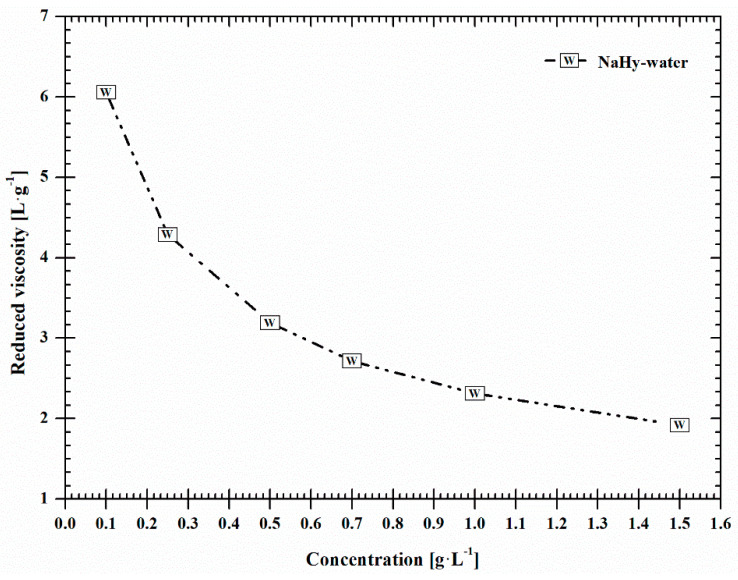
Reduced viscosity of NaHy in water.

**Table 1 ijms-22-01932-t001:** Tracer diffusion coefficients, ^app^*D*^0^_1T_, for sodium hyaluronate (NaHy) 0.1% ((Mw = 1.8 MDa)) (component 1) in aqueous solutions containing different salts (component 2) at 0.01 mol dm^−3^ and 0.1 mol dm^−3^, and the respective standard deviations of the mean ^1^, *S_D_*, and at 25 °C.

	Cation (K_C_ or CH_C_) ^2^	Anion (K_A_ or CH_A_) ^2^	^app^*D*^0^_1T_ ± *S_D_*/(10^−9^m^2^ s^−1^) (0.01 mol dm^−3^)	(Δ^app^*D*^0^_1T_/*D*^0^) % ^3^	^app^*D*^0^_1T_ ± *S_D_*/(10^−9^m^2^ s^−1^) (0.1 mol dm^−3^)	(Δ^app^*D*^0^_1T_/*D*^0^) % ^3^
MgSO_4_	(Mg^2+^, K_c_)	(SO_4_^2^^−^, K_A_)	0.609 ± 0.004	−54.3	0.715 ± 0.005	−46.4
NaSCN	(Na^+^, K_C_)	(SCN^−^, CH_A_)	0.858 ± 0.004	−35.6	1.421 ± 0.004	6.6
Na_2_SO_4_	(Na^+^, K_C_)	(SO_4_^2^^−^, K_A_)	0.860 ± 0.019	−35.5	1.016 ± 0.003	−23.8
NH_4_SCN	(NH_4_^+^, CH_C_)	(SCN^−^, CH_A_)	1.080 ± 0.020	−18.9	1.595 ± 0.007	19.7
NaCl	(Na^+^, K_C_)	(Cl^−^, CH_A_)	1.090 ± 0.020	−25.7	1.435 ± 0.008	7.6
LiCl	(Li^+^, K_C_)	(Cl^−^, CH_A_)	1.360 ± 0.003	2.0	1.351 ± 0.005	1.3
(NH_4_)_2_SO_4_	(NH_4_^+^, CH_C_)	(SO_4_^2^^−^, K_A_)	1.109 ± 0.008	−16.8	1.313 ± 0.004	−1.5

^1^ Averaged result for *n* = 3 experiments. Standard uncertainties *u* are: u_r_(*c*) = 0.03; *u*(*T*) = 0.01 °C and *u*(*P*) = 2.03 kPa. ^2^ K_c_ or CH_c_, and Ka or CH_A_, means kosmotropic cation (or chaotropic cation), and kosmotropic anion (or chaotropic anion). ^3^ (Δ^app^*D*^0^_1T_/*D*^0^) represents the deviations between the apparent tracer diffusion coefficients of NaHy in aqueous electrolytic solutions at 0.01 or 0.1 mol dm^−3^, and the limiting diffusion coefficients of the aqueous solutions of NaHy at the same temperature *D*^0^ = 1.333 × 10^−9^ m^2^ s^−1^ [18].

**Table 2 ijms-22-01932-t002:** Tracer diffusion coefficients, ^app^*D*^0^_1T_, for NaHy 0.1% ((Mw = 124 kDa)) (component 1) in aqueous solutions containing different salts (component 2) at 0.01 mol dm^−3^ and 0.1 mol dm^−3^, and the respective standard deviations of the mean ^1^, *S_D_*, and at 25 °C.

	Cation (K_C_ or CH_C_) ^2^	Anion (K_A_ or CH_A_) ^2^	^app^*D*^0^_1T_ ± *S_D_*/(10^−9^m^2^ s^−1^) (0.01 mol dm^−3^)	(Δ^app^*D*^0^_1T_/*D*^0^) % ^3^	^app^*D*^0^_1T_ ± *S_D_*/(10^−^^9^m^2^ s^−^^1^) (0.1 mol dm^−3^)	(Δ^app^*D*^0^_1T_/*D*^0^) % ^3^
MgSO_4_	(Mg^2+^, K_c_)	(SO_4_^2^^−^, K_A_)	0.709 ± 0.002	−31.6	0.754 ± 0.005	−27.2
NaSCN	(Na^+^, K_C_)	(SCN^−^, CH_A_)	1.136 ± 0.002	9.6	1.325 ± 0.010	27.8
Na_2_SO_4_	(Na^+^, K_C_)	(SO_4_^2^^−^, K_A_)	0.868 ± 0.010	−16.2	1.075 ± 0.020	3.8
NH_4_SCN	(NH_4_^+^, CH_C_)	(SCN^−^, CH_A_)	1.380 ± 0.010	33.2	1.563 ± 0.001	50.9
NaCl	(Na^+^, K_C_)	(Cl^−^, CH_A_)	0.947 ± 0.002	−8.6	1.426 ± 0.005	37.6
LiCl	(Li^+^, K_C_)	(Cl^−^, CH_A_)	1.282 ± 0.005	23.4	1.228 ± 0.005	18.5
(NH_4_)_2_SO_4_	(NH_4_^+^, CH_C_)	(SO_4_^2^^−^, K_A_)	1.200 ± 0.005	15.8	1.490 ± 0.004	43.8

^1^ Averaged result for *n* = 3 experiments. Standard uncertainties *u* are: *u*_r_(*c*) = 0.03; *u*(*T*) = 0.01 °C and *u*(*P*) = 2.03 kPa. ^2^ K_C_ or CH_C_, and K_A_ or CH_A_, means kosmotropic cation (or chaotropic cation), and kosmotropic anion (or chaotropic anion). ^3^ (Δ^app^*D*^0^_1T_/*D*^0^) represents the deviations between the apparent tracer diffusion coefficients of NaHy in aqueous electrolytic solutions at 0.01 or 0.1 mol dm^−3^, and the limiting diffusion coefficients of the aqueous solutions of NaHy at the same temperature *D*^0^ = 1.036 × 10^−9^ m^2^ s^−1^ [18] obtained by extrapolated values obtained from the *D* least-squares for three injections of NaHy in water (that is, for *c* = 0.1, 0.5 and 1 g dm^−3^, we obtained *D* = 1.014, 0.859 and 0.724 × 10^−9^m^2^ s^−1^).

**Table 3 ijms-22-01932-t003:** Behaviour of NaHy in salt solutions observed by measurements of apparent tracer diffusion coefficients ^app^*D*^0^_1T_ and reduced viscosity *η*_red_ (NaHy, Mw = 124 kDa, ionic strength 0.1 mol L^−1^) at 25 °C. Arrows indicate upward and downward curvature of viscosity dependences at lowest concentrations Parameter.

	Salts Behaviour						
^app^*D*^0^_IT_ × 10^−9^ (m^2^·s^−1^)	MgSO_4_	Na_2_SO_4_	LiCl	NaSCN	NaCl	(NH_4_)_2_SO_4_	NH_4_SCN
	**KK**	**KK**	**KCH**	**KCH**	**KCH**	**CHK**	**CHCH**
*η*_red_ water (L·g^−1^)	NH_4_SCN	(NH_4_)_2_SO_4_	NaSCN	NaCl	Na_2_SO_4_	LiCl	MgSO_4_
	**CHCH**	**CHK**	**KCH**	**KCH**	**KK**	**KCH**	**KK**
*η*_red_ NaHy at 0.1 g L^−1^ (L·g^−1^)	NaCl	NH_4_SCN	(NH_4_)_2_SO_4_	LiCl	Na_2_SO_4_	MgSO_4_	NaSCN
	↓	**↓**	↓	↑	↑	↑	↑
	**KCH**	CHCH	**CHK**	**KCH**	**KK**	**KK**	**KCH**
*η*_red_ NaHy at 1.5 g L^−1^(L·g^−1^)	MgSO_4_	NaCl	NH_4_SCN	LiCl	NaSCN	(NH_4_)_2_SO_4_	Na_2_SO_4_
	**KK**	**KCH**	**CHCH**	**KCH**	**KCH**	**CHK**	**KK**

## Data Availability

Data is contained within the article.

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
