# Peer review of "Effect of Hofmeister Ions on Transport Properties of Aqueous Solutions of Sodium Hyaluronate"

_ijms, 2021, doi:10.3390/ijms22041932_

Round 1
Reviewer 1 Report
Ref: ijms-1104037
Title: Effect of Hofmeister ions on transport properties of aqueous solutions of sodium hyaluronate
Journal: International Journal of Molecular Sciences
In this manuscript tracer diffusion coefficients by Taylor dispersion technique were measured and viscosity measurements were carried out to study the influence of several salts on the transport behavior of sodium hyaluronate. The results of this study could be applied in modelling diffusion in pharmaceutical and engineering fields. Introduction and discussion are supported by a fair number of bibliographic references which include both "historical" studies and very recent papers. The study was dealt rigorously and the results are clearly and conclusively explained.
There are only a few oversights in the manuscript. I would recommend publishing on International Journal of Molecular Sciences after a very minor revision.
Minor suggestions are listed below:
- In Table 1, referring to NaCl, “(SCN- , CHA)” must be corrected as “(Cl-, CHA)”.
- In Table 2, referring to NaCl, “(SCN- , CHA)” must be corrected as “(Cl-, CHA)”.
- Line 289, “the Na+”, where + must be superscript.
- Line 525, reference 40, subscripts in “MgCl2, MgSO4 ...”.
Author Response
All changes suggested by the Reviewer have been done (see highlighted version)
Reviewer 2 Report
The authors evaluate the behavior of the diffusion coefficient and the viscosity of aqueous solutions of sodium hyaluronate, at different ion concentrations of the Hofmeister Series.
The manuscript is very complete, the quality of the experimental data is good, and the analysis of the results is well founded. The conclusions are well supported and represent a good example of the quality of the manuscript.
One of the most striking results is the salting-in effect of some kosmotropic ions (structure-forming) and the salting-out effect of the disturbing ions of the water structure and their effect on the diffusion coefficient. Finally the research work is very good, and I recommend publishing it without any changes.
Author Response
We are grateful for such positive comments.